# Hydrodynamics, Hydrochemistry, and Stable Isotope Geochemistry to Assess Temporal Behavior of Seawater Intrusion in the La Yarada Aquifer in the Vicinity of Atacama Desert, Tacna, Peru

Alissa Vera [1], Edwin Pino-Vargas [1], Mahendra P. Verma [2,*], Samuel Chucuya [3], Eduardo Chávarri [4], Miguel Canales [4], Juan Antonio Torres-Martínez [5], Abrahan Mora [6] and Jürgen Mahlknecht [5]

1   Departamento de Ingenieria Civil, Universidad Nacional Jorge Basadre Grohmann, Tacna 23000, Peru; averam@unjbg.edu.pe (A.V.); epinov@unjbg.edu.pe (E.P.-V.)
2   Geociencia, Universidad Politécnica de Nochixtlán "Abraham Castellanos", Oaxaca 69600, Mexico
3   Departamento de Geologia-Geotecnia, Universidad Nacional Jorge Basadre Grohmann, Tacna 23000, Peru; schucuyam@unjbg.edu.pe
4   Departamento de Recursos Hídricos, Universidad Nacional Agraria La Molina, Lima 15012, Peru; echavarri@lamolina.edu.pe (E.C.); miguelcanales@lamolina.edu.pe (M.C.)
5   Escuela de Ingeniería y Ciencias, Tecnológico de Monterrey, Campus Monterey, Monterrey 64849, Mexico; antoniotorres@tec.mx (J.A.T.-M.); jurgen@tec.mx (J.M.)
6   Escuela de Ingeniería y Ciencias, Tecnológico de Monterrey, Campus Puebla, Puebla 72453, Mexico; abrahanmora@tec.mx
*   Correspondence: mpv55.mx1@gmail.com

**Abstract:** The La Yarada aquifer is the primary water resource for municipal, irrigation, and industrial uses in the semi-arid Tacna, Peru. Presently, over-pumping has caused severe groundwater management problems, including the abandonment of saline water wells. This study presents multivariate analysis and chemical–isotopic trends in water to investigate seawater intrusion and hydrogeological processes affecting water quality. The chemical and isotopic analysis of water samples, collected in two campaigns in the dry (August 2020) and wet (November 2020) seasons, together with the 1988 data, were evaluated with a mixing model, cluster, and factor analysis. The hydrochemical and isotopic mixing model suggested the formation of a wedge with 20% seawater intrusion. The heterogeneity of piezometric map isolines corroborates the wedge formation associated with the groundwater movement. The spatial distributions of factors, FA1 and FA2, suggest two processes of seawater front movement: dispersion (diffusion) of chemical elements and different types of water mixing, respectively. At the edge of the La Yarada aquifer, the water head was relatively low, permitting seawater and freshwater mixing. On the other hand, along the sea-land boundary, the water head of the La Yarada aquifer was relatively high, avoiding seawater and freshwater mixing; however, the chemical species were migrating from the seawater to the groundwater due to the diffusion processes. The cluster 4 samples are in the region corresponding to the isotopic mixing process represented by the FA2, while cluster 4 describes the chemical diffusion process represented by the FA2. Thus, the integrated approach is helpful to assess the seawater intrusion mechanisms in coastal aquifers in a semi-arid region.

**Keywords:** coastal aquifer; seawater intrusion; salinization; isotopic signature

## 1. Introduction

In coastal aquifers, seawater intrusion is a challenging problem affecting groundwater management [1]. However, the situation is more complex in the arid and semi-arid areas, resulting in aquifer heads declination, groundwater quality deterioration, lower crop yields, and ecosystem degradation [2]. With rapid industrial growth and agriculture intensification, groundwater sustainability has become a significant issue in Tacna, Peru [3,4]. The seawater

infiltration into the aquifer due to overexploitation with high pumping wells [5,6] had forced the abandonment of wells in the region with increased salinity [7,8]. The encroachment of saline water into fresh groundwater has been a subject of extensive study for well over a century [9]. Since 1920, various graphic and statistical techniques have facilitated the homogeneous grouping of groundwater [10,11]. Multivariate statistical methods, cluster analysis (CA), and factor analysis (FA) effectively manipulate, interpret, and represent groundwater pollutants and geochemistry data [12,13].

Villegas et al. [14] used the principal component analysis to evaluate the dissolution and cation-exchange processes controlling the groundwater chemistry in the Uraba aquifer. Black et al. [15] assessed the thermal water origin in the Dublin carboniferous basin (Ireland) by implementing CA and FA. Werner and Simmons [16] illustrated the first-order assessment of seawater intrusion through a simple conceptual framework, classifying aquifers as flux-controlled and head-controlled systems. Binda et al. [17] argued for applying an integrated interdisciplinary approach and illustrated it by evaluating potentially toxic element sources in a mountainous watershed. The same procedure defined the natural and anthropogenic contributions in the winter snow of the Dolomites, Italy [18].

The present study implemented the integrated interdisciplinary approach [17] to characterize the spatiotemporal variations of water level and salt content in the La Yarada coastal aquifer in the vicinity of the Atacama Desert, Tacna, Peru. The study compiled the hydrodynamic, hydrochemical, and isotopic data ($\delta^{18}$O and $\delta^2$H) from the literature. Water samples of ice, rivers, springs, and groundwater wells in the basin were collected in two campaigns. A multivariate statistical analysis was performed to characterize the dominant factors to understand the mechanism of seawater intrusion in the aquifer. Likewise, an isotope–hydrochemical mixing method was employed to identify the salinization processes due to seawater intrusion in the aquifer.

## 2. Study Area

### 2.1. Location and Climate

The Atacama Desert, a cold arid region in northern Chile, is approximately 1100 km long from north to south, and the north part of the desert is in Peru. Figure 1 shows the study area of the Caplina hydrographic basin, which encompasses a delta that caused the formation of the La Yarada coastal aquifer. Desertification started 14 Ma during the global climatic desiccation [19]. Similarly, the Barroso mountain range separates the Caplina and Maure basins [20]. The coast and the entire western slope of the mountain range are in a rainy region of around 400 mm. The annual rainfall in the Caplina basin is only up to 10 mm, except for the years 2019 and 2020 [21].

The main economic activity in the area is agriculture, covering more than 40,000 ha cultivated on the surface of the La Yarada aquifer in 2020 [21,22]. Consequently, the groundwater has a severe imbalance problem in the coastal zone to meet the population irrigation and domestic use demand [20]. The principal causes of the aquifer's progressive salinization process are quality deterioration due to seawater intrusion and depletion of the aquifer [22,23]. Consequently, there is a crisis of governability and governance in groundwater use [23]. The excessive and uncontrolled groundwater pumping fulfilled the local water resources demand in the last century [19,21]. In 2018, illegal pumping wells exploited more than 160 Hm$^3$ per year [22], far exceeding the aquifer capacity.

### 2.2. Hydrogeological Settings

The La Yarada aquifer is mainly of quaternary alluvial origin. Its shape is a rectangular polygon with a flat bottom and steep and abrupt flanks. Downstream, the ejected cone of the Caplina river (Peru) constitutes a physiographic unit that begins in the Magollo gorge and progressively widens downward in a delta unit, reaching the beach line [24]. The aquifer is delimited from Calientes to the beach line from the northeast to the southwest by rocky outcrops with incipient wind cover and volcanic ash deposits [25].

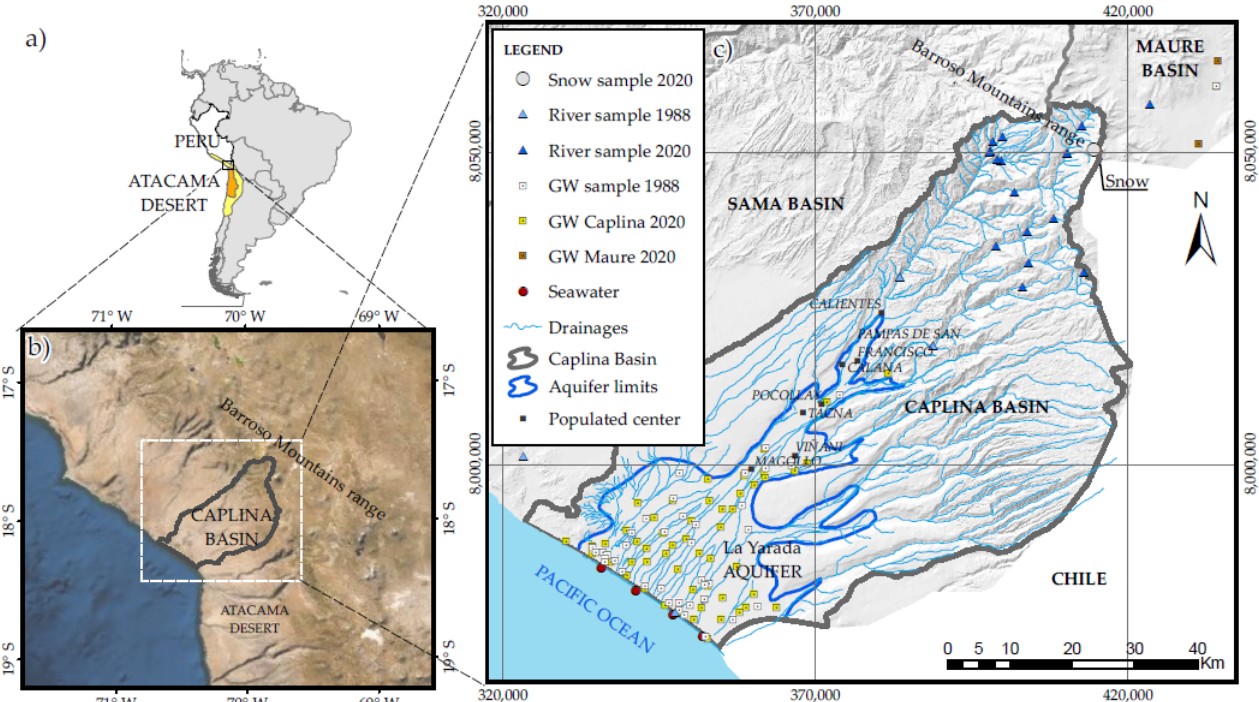

**Figure 1.** Location of the study area. (**a**) Atacama Desert in Chile and Peru, (**b**) Barroso Mountain range, a separation boundary between Caplina and Maure basins, (**c**) La Yarada coastal aquifer in the Caplina basin.

During the last two decades, the water level has declined due to groundwater overuse, and consequently, there was seawater intrusion in the aquifer. In the coming years, the water level will continue to drop from 0.23 to 0.38 m/year, and saline intrusion will be five times higher (56–59 hm$^3$/year), increasing groundwater quality deterioration [23,26].

## 3. Materials and Methods

### 3.1. Monitoring Network and Sampling

The hydrochemical and isotopic monitoring network, distributed in the Caplina basin, was first established in an IAEA project in 1988 [27]. In this study, the first campaign was performed in August 2020 (dry season), while the second was in November 2020 (wet season) [28,29]. One hundred sampling points were selected, distributed in three basins: 86 points from the Caplina basin, 9 points from the NE Maure basin, and 5 points in the NW Sama basin for the chemical and isotopic analyses (Figure 1). Of the total samples, 76 are for agriculture use, 9 for domestic use, 5 for cattle-ranch use, 3 for urban use, 1 for industrial use, and 6 for no use due to high contamination.

Each sample was conducted in 4 days from the highest part of the Maure basin at 4500 m above sea level (masl) to the beach at 0 masl by two brigades. The sample points of the Maure and Sama basins were selected to observe the influence of interaction or contribution from the aquifers to the La Yarada aquifer of the Caplina basin.

For the present study, we also compiled the isotopic data of 44 groundwater and three from the Caplina and Uchusuma rivers from the report of the IAEA project [24].

### 3.2. Analysis and Data Quality

Cation (Ca$^{2+}$, Mg$^{2+}$, Na$^+$, K$^+$) analysis of water samples was conducted by inductively coupled plasma mass spectrometry (ICP-MS) following the method ISO 17294-2. The anions (SO$_4^{2-}$, Cl$^-$, NO$_3^-$, F$^-$) were analyzed on the ion chromatograph, using EPA methods 300.0. The carbonic species (HCO$_3^-$) and alkalinity were determined by the acid-base titration method. The $\delta^{18}$O and $\delta^2$H of water samples were measured by the analyzer ABB-LGR model 912-0008 with analytical precision $\pm$ 0.1‰ and $\pm$ 1.0‰, respectively.

### 3.3. Statistical Interpretation Procedure

The univariate, bivariate, and multivariate analyses of the chemical and isotopic data were performed with the computer code written during this study in Python. The following 14 variables were considered: electrical conductivity (EC), pH, total dissolved solids (TDS), $Ca^{2+}$, $Mg^{2+}$, $Na^+$, $K^+$, $HCO_3^-$, $SO_4^{2-}$, $Cl^-$, $NO_3^-$, $F^-$, $\delta^{18}O$, and $\delta^2H$ to identify chemical groups in groundwater.

### 3.4. Groundwater and Seawater Mixing Model

A mixing model was proposed to quantify the seawater component in the Caplina basin. It consists of establishing the chemical and isotopic compositions of end members: (1) local fresh groundwater ($X_{gnd}$) and (2) seawater ($X_{sea}$) [30]. The fraction of seawater component ($\gamma$) in each groundwater well is calculated as

$$\gamma = \frac{X_{mix} - X_{gnd}}{X_{sea} - X_{gnd}} \tag{1}$$

where $X$ is the concentration of chemical ($Cl^-$, $Na^+$) and isotopic ($\delta^{18}O$) species in the saline groundwater samples. The subscript (mix) indicates the mixture component (i.e., the La Yarada aquifer) groundwater.

Figure 2 presents the conceptual model of the negative effect on the hydraulic barriers [31] and geometry of the freshwater–seawater contact after 13 years of operation of extraction wells [32]. On defining the end members (i.e., local fresh groundwater and seawater), Equation (1) is used to calculate the seawater fraction in each groundwater well in the aquifer. Thus, the mixing model permits in defining the fresh–seawater contact geometry.

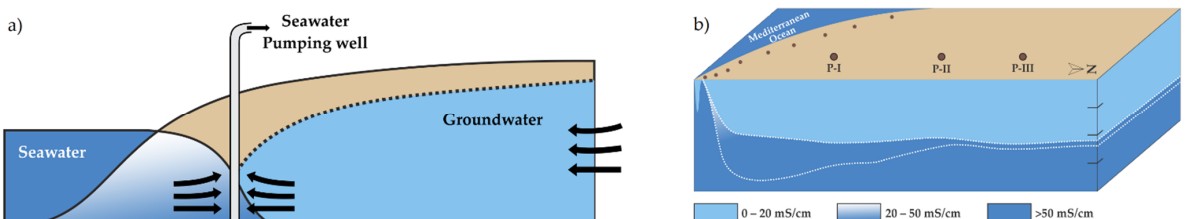

**Figure 2.** (**a**) Conceptual model of the negative effect on the hydraulic barriers, modified after [31] and (**b**) geometry of the freshwater–seawater contact, predicted from the mixing model [32].

## 4. Results and Discussion

### 4.1. Spatial and Temporal Univariate Trends in Groundwater Chemistry

Table 1 presents the data statistics of chemical and isotopic analyses of samples collected in 2020. The physical chemistry of saline groundwater has more significant variability in EC, which is related to TDS. Figure 3 shows the spatial and temporal distribution of $Na^+$ and $Cl^-$ in the aquifer for 1998 and 2020. These parameters are generally conservative during the mixing of different types of water [30]. The iso-value curves indicate that the high $Na^+$ and $Cl^-$ concentration zones had evolved from W to SW along the aquifer edge in contact with seawater.

**Table 1.** Data statistical of 14 physicochemical and isotopic parameters in 2020, sampled in the dry (August 2020) and wet (November 2020) seasons.

| Statistics | CE | pH | TDS | $Ca^{2+}$ | $Mg^{2+}$ | $Na^+$ | $K^+$ | $HCO_3^-$ | $SO_4^{2-}$ | $Cl^-$ | $NO_3^-$ | $F^-$ | $\delta^{18}O$ | $\delta^2H$ |
|---|---|---|---|---|---|---|---|---|---|---|---|---|---|---|
| | μS/cm | | | | | | mg/L | | | | | | ‰ | |
| | | | | | | Dry Season (August 2020) | | | | | | | | |
| Mean | 1772.8 | 7.6 | 1248.6 | 164.9 | 34.3 | 161.8 | 17.4 | 87.8 | 410.3 | 362.1 | 3.6 | 0.2 | −12.1 | −88.0 |
| Minimum | 25.5 | 5.8 | 63.5 | 10.7 | 0.8 | 47.9 | 1.5 | 53.4 | 45.4 | 65.5 | 0.02 | 0.1 | −13.5 | −96.2 |
| Maximum | 8190.0 | 8.7 | 5594.0 | 874.0 | 223.0 | 872.0 | 51.7 | 196.0 | 1030.0 | 2417.0 | 20.4 | 0.6 | −6.7 | −49.9 |
| | | | | | | Wet Season (November 2020) | | | | | | | | |
| Mean | 2075.1 | 7.6 | 1381.1 | 185.6 | 35.2 | 166.5 | 15.1 | 86.6 | 394.7 | 369.9 | 4.1 | 0.2 | −12.2 | −86.4 |
| Minimum | 653.0 | 7.0 | 425.0 | 12.1 | 0.9 | 51.8 | 2.8 | 46.2 | 42.8 | 70.2 | 0.6 | 0.1 | −13.8 | −95.8 |
| Maximum | 8290.0 | 8.3 | 5918.0 | 899.6 | 209.1 | 871.0 | 44.3 | 170.0 | 969.0 | 2266.0 | 24.1 | 0.6 | −4.7 | −31.2 |

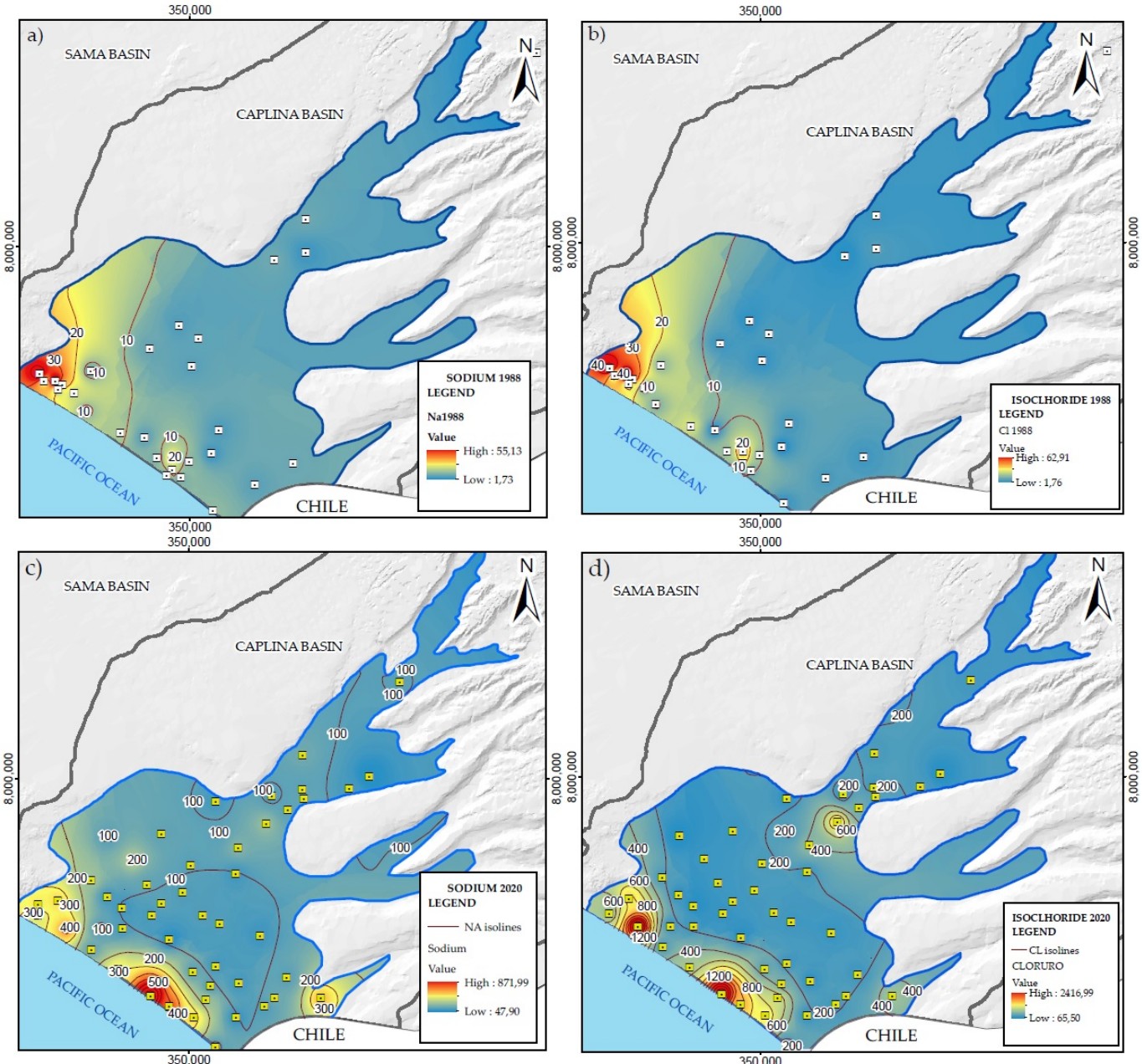

**Figure 3.** Iso-value plot in the groundwater of the La Yarada aquifer (this study) for 1988: (**a**) Na$^+$, (**b**) Cl$^-$, and for 2020: (**c**) Na$^+$, (**d**) Cl$^-$. The high concentration zones of Na$^+$ and Cl$^-$ (i.e., seawater intrusion) have moved from W to SW along the sea-land boundary during the period.

Similarly, the saline seawater front is moving slowly inside the coastal area. The phreatic level inside the Caplina basin was decreasing due to extensive pumping up to 2010. However, high precipitation in 2019 and 2020 was favorable for improving the phreatic level profile [33]; however, there was no significant change in the chemical and isotopic compositions of the La Yarada groundwater.

### 4.1.1. Stable Isotope of Caplina Basin

Figure 4 shows the isotopic values of different types of waters in the Craig diagram [34]. Ninety percent of the water samples lie around the GMWL, and the data were fitted to the LMWL, $\delta^2H = 7.06\,\delta^{16}O + 0.97$, with R$^2$ = 0.97 in 2020. There is a high scattering in the values of 1988; however, the values are around the GMWL.

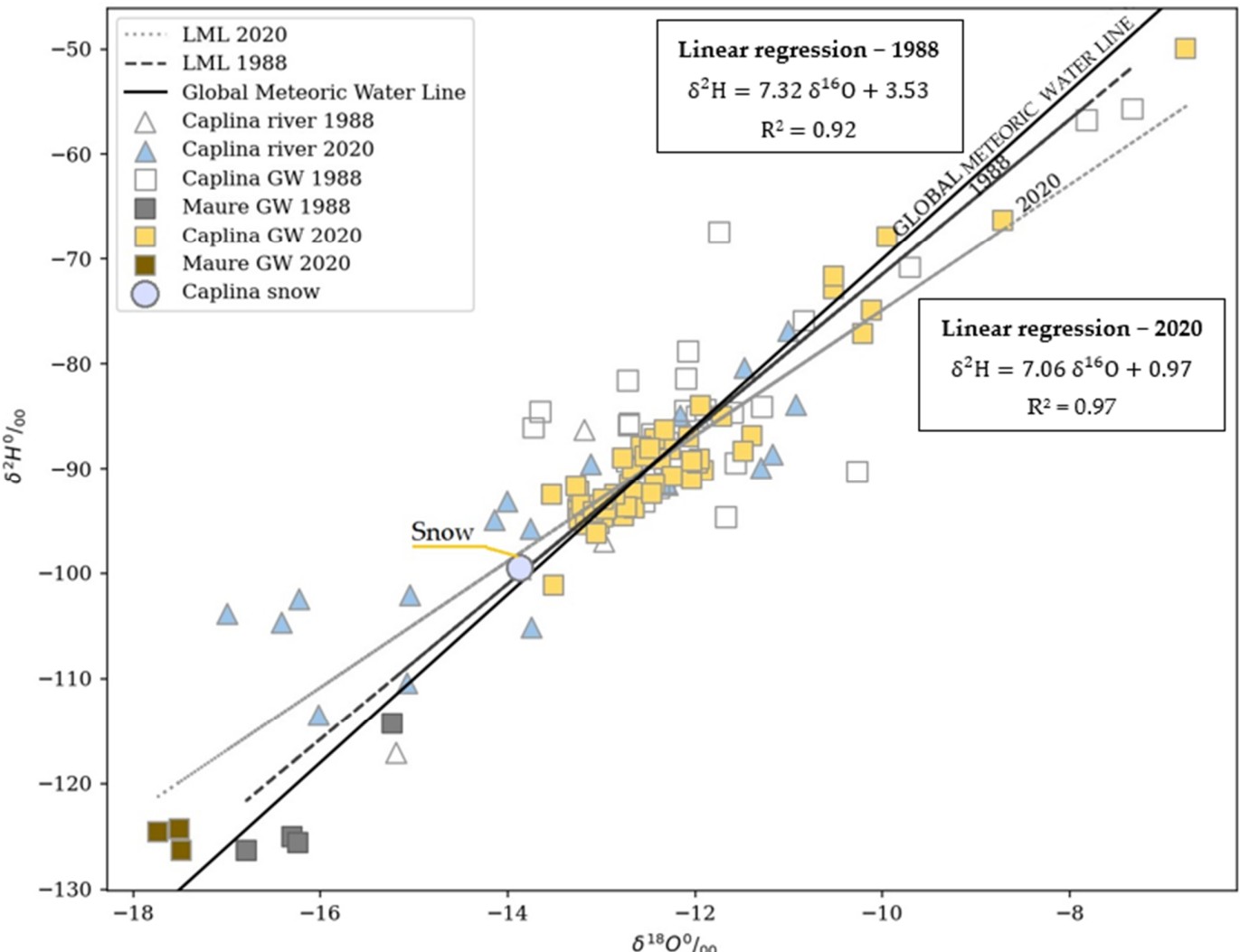

**Figure 4.** Distribution of $\delta^{18}O$ and $\delta^2H$ values of samples collected at the study area with the global meteoric water line (GMWL). The local meteoric water lines (LMWL) for 1988 and 2020 were plotted with respective equations. The isotopic values of Caplina and Maure Basin are on the trend line.

The snow samples ($\delta^{18}O = -13.9‰$ and $\delta^2H = -99.6‰$) were collected at the elevation (Lat. = 17°37′ and Long. = 69°48′, Elev. = 5164 masl). The highest peak of the mountain is at the elevation (Elev. = 5800 masl). So, the isotopic composition of rainwater at the mountain range may still be lighter and close to that of the Maure aquifer. The Caplina basin is probably recharged by the river waters and evaporated during the river flow trajectory. The isotopic values of Caplina and Maure Basin are on the trend line. The isotopic values of snow suggest the Barroso Mountain range as a separation boundary for the basins. The groundwaters of the La Yarada aquifer are the recharge of the evaporated mountain rain waters.

### 4.1.2. Isotopic Composition Interpolation

The isotopic composition of groundwater in arid regions may be different from the composition of the recharge zone [28]. Figure 5a presents isotopic composition iso-lines that separate the isotopic system of the Sama basin in 1988 in the littoral part; however, the iso-line of −11‰ stands out at 2 km from the coastline and −12‰ at 5 km, generating an isotopic enrichment anomaly due to seawater influence in 2020 (Figure 5b).

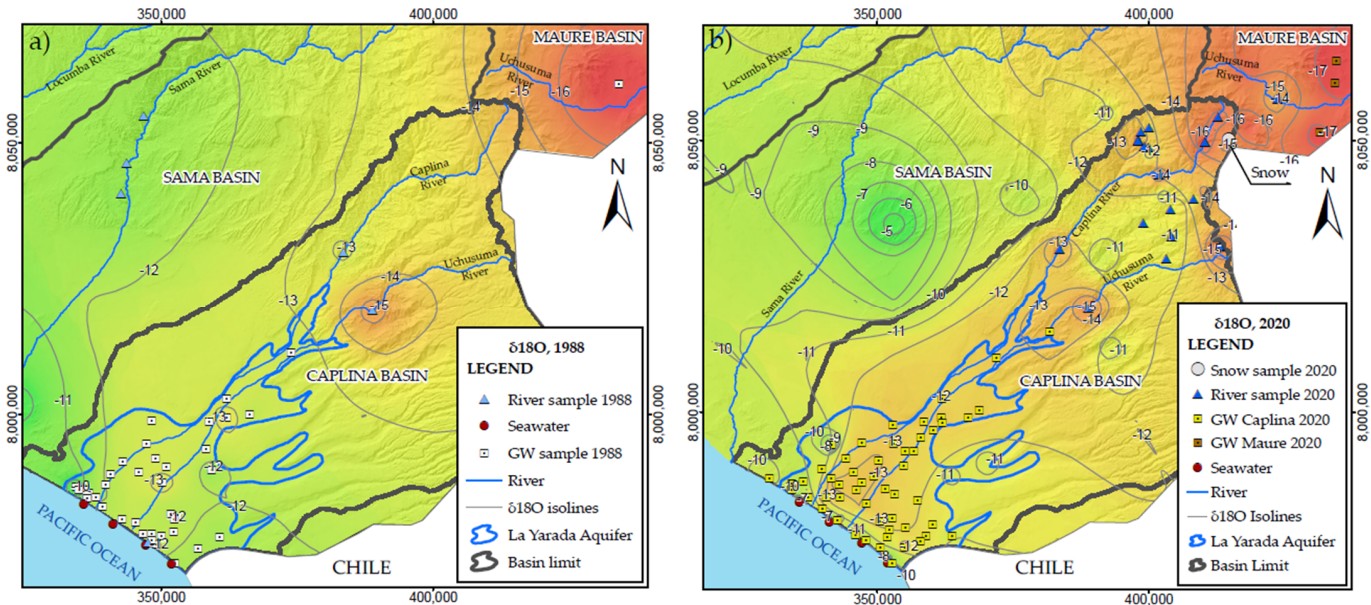

**Figure 5.** (**a**) Spatial distribution of $\delta^{18}O$ in the Caplina, Sama, and Maure hydrological basins in 1988. (**b**) Update of the isotopic composition in the aquifer for the year 2020.

### 4.2. Statistical Correlation

Table 2 shows the correlation matrix among the 14 variables. The correlation coefficients between the physicochemical parameters and isotopes of $\delta^2H$ and $\delta^{18}O$ have low values except for fluorine. Likewise, the strong positive correlations in pairs TDS and $Cl^-$ (0.98), TDS and $Ca^{2+}$ (0.92), TDS and $Mg^{2+}$ (0.93) reflect the dependence of salinity increase with the primary ions of seawater $Cl^-$ (0.98), $Ca^{2+}$ (0.92), and $Mg^{2+}$. However, the increase in TDS is not necessarily due to seawater intrusion. It is crucial to consider the solid correlation for salts corresponding to seawater [35,36], highlighting the exclusive non-correspondence of magnesium and calcium coming from calcareous rocks. Similarly, the weak correlations of $F^-$, $NO_3^-$, and $HCO_3^-$ with electrical conductivity (EC) and TDS reveal a low contribution in groundwater salinity compared to the major ions $Cl^-$, $Ca^{2+}$, and $Mg^{2+}$.

**Table 2.** Correlation coefficient table for 14 parameters of water sample in 2020. EC stands for electrical conductivity ($\mu$S/cm), TDS is total dissolved solids (mg/L). The significant correlation values are marked as bold.

|  | EC | pH | TDS | $Ca^{2+}$ | $Mg^{2+}$ | $Na^+$ | $K^+$ | $HCO_3^-$ | $SO_4^{2-}$ | $Cl^-$ | $NO_3^-$ | $F^-$ | $\delta^{18}O$ | $\delta^2H$ |
|---|---|---|---|---|---|---|---|---|---|---|---|---|---|---|
| **EC** | 1.00 | −0.40 | **0.70** | **0.78** | **0.73** | 0.45 | 0.65 | 0.25 | 0.50 | 0.66 | 0.35 | 0.31 | 0.10 | 0.12 |
| pH | −0.40 | 1.00 | −0.52 | −0.62 | −0.55 | −0.29 | −0.30 | −0.47 | −0.52 | −0.47 | −0.51 | 0.22 | 0.46 | 0.48 |
| TDS | **0.70** | −0.52 | 1.00 | **0.92** | **0.93** | **0.86** | **0.84** | 0.19 | **0.71** | **0.98** | 0.43 | 0.27 | 0.03 | 0.08 |
| $Ca^{2+}$ | **0.78** | −0.62 | **0.92** | 1.00 | **0.97** | 0.60 | **0.76** | 0.24 | **0.72** | **0.88** | 0.48 | 0.14 | −0.11 | −0.09 |
| $Mg^{2+}$ | **0.73** | −0.55 | **0.93** | **0.97** | 1.00 | 0.66 | 0.82 | 0.14 | **0.77** | **0.89** | 0.33 | 0.11 | −0.13 | −0.07 |
| $Na^+$ | 0.45 | −0.29 | **0.86** | 0.60 | 0.66 | 1.00 | **0.78** | 0.16 | 0.52 | **0.87** | 0.30 | 0.43 | 0.24 | 0.32 |
| $K^+$ | 0.65 | −0.30 | **0.84** | **0.76** | **0.82** | **0.78** | 1.00 | 0.06 | **0.70** | **0.80** | 0.23 | 0.28 | 0.10 | 0.21 |
| $HCO_3^-$ | 0.25 | −0.47 | 0.19 | 0.24 | 0.14 | 0.16 | 0.06 | 1.00 | 0.28 | 0.13 | 0.53 | 0.03 | −0.06 | −0.13 |
| $SO_4^{2-}$ | 0.50 | −0.52 | **0.71** | **0.72** | **0.77** | 0.52 | 0.70 | 0.28 | 1.00 | 0.58 | 0.27 | −0.16 | −0.40 | −0.31 |
| $Cl^-$ | 0.66 | −0.47 | **0.98** | **0.88** | **0.89** | **0.87** | **0.80** | 0.13 | 0.58 | 1.00 | 0.40 | 0.35 | 0.12 | 0.17 |
| $NO_3^-$ | 0.35 | −0.51 | 0.43 | 0.48 | 0.33 | 0.30 | 0.23 | 0.53 | 0.27 | 0.40 | 1.00 | 0.15 | −0.02 | −0.11 |
| $F^-$ | 0.31 | 0.22 | 0.27 | 0.14 | 0.11 | 0.43 | 0.28 | 0.03 | −0.16 | 0.35 | 0.15 | 1.00 | **0.74** | **0.75** |
| $\delta^{18}O$ | 0.10 | 0.46 | 0.03 | −0.11 | −0.13 | 0.24 | 0.10 | −0.06 | −0.40 | 0.12 | −0.02 | 0.74 | 1.00 | 0.96 |
| $\delta^2H$ | 0.12 | 0.48 | 0.08 | −0.09 | −0.07 | 0.32 | 0.21 | −0.13 | −0.31 | 0.17 | −0.11 | **0.75** | **0.96** | 1.00 |

### 4.3. Multivariate Analysis

4.3.1. Cluster Analysis

The cluster analysis is a statistical tool to classify the principal data groups according to a Euclidean distance dissimilarity measure [36]. According to their parameters, the sample classifications are known as Q mode classifications. The Ward method uses Euclidean distances to measure similarity and has a small distortion effect in space [12,13]. The 14 chemical parameters (EC, pH, TDS, $Ca^{2+}$, $Mg^{2+}$, $Na^+$, $K^+$, $HCO_3^-$, $SO_4^{2-}$, $Cl^-$, $NO_3^-$, $F^-$, $\delta^{18}O$, and $\delta^2H$) of 53 groundwater samples of the La Yarada aquifer were grouped into four clusters of hydrogeochemical facies according to their similarity in chemical signature (Figure 6).

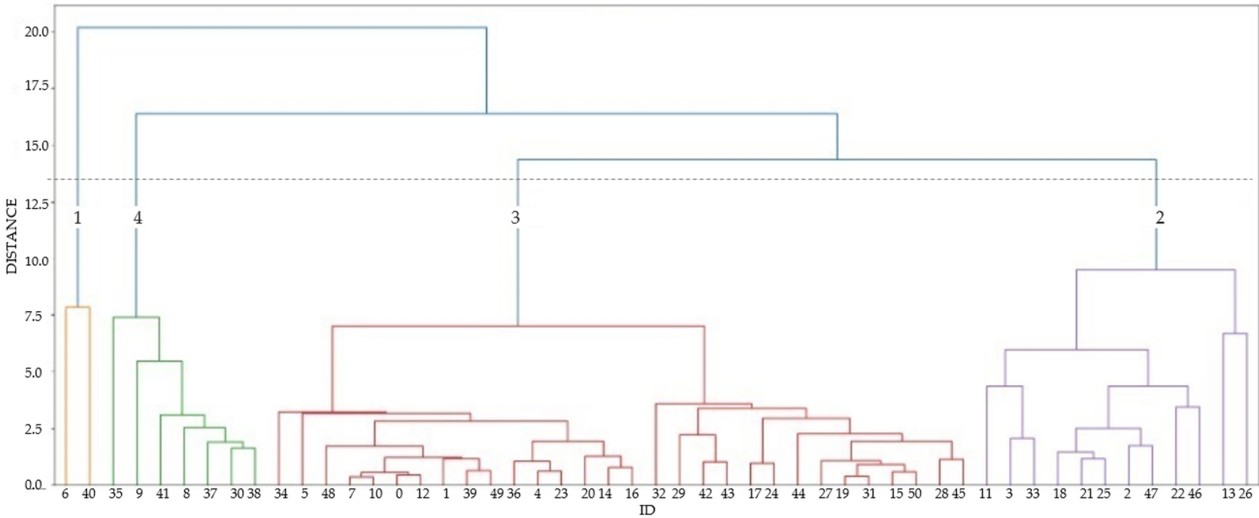

**Figure 6.** Dendrogram derived using 14 parameters of 53 groundwater samples the log-transformed and standardized (z-score) water chemistry and isotopic ratios dataset for 2020 sampling campaign. A dashed horizontal line is used to define four groups of water samples.

Table 3 explains the relative abundance of ions. $Cl^-$ and $SO_4^{2-}$ are the dominant ions in all clusters except for cluster 2, which shows the domination of $Cl^-$ and $Na^+$. A relationship (<0.05) of the $Na^+/Cl^-$ ratio reveals the influence of very salty water from seawater intrusion [36]. Figure 7a presents the spatial distribution of the chemical signature of the four clusters. Cluster 1 is parallel to the coastline and has high TDS, confirming the high salinity of the coastline.

**Table 3.** Relative ion abundance of the four clusters.

| Cluster | Relative Abundance | | | | | | | | | | | | | |
|---------|----|----|----|----|----|----|----|----|----|----|----|----|----|----|
| 1 | $Cl^-$ | >> | $SO_4^{2-}$ | >> | $Ca^{2+}$ | > | $Na^+$ | >> | $Mg^{2+}$ | > | $HCO_3^-$ | > | $K^+$ | > | $NO_3^-$ |
| 2 | $Cl^-$ | >> | $Na^+$ | > | $SO_4^{2-}$ | >> | $Ca^{2+}$ | > | $HCO_3^-$ | >> | $K^+$ | > | $Mg^{2+}$ | > | $NO_3^-$ |
| 3 | $SO_4^{2-}$ | >> | $Cl^-$ | > | $Ca^{2+}$ | > | $Na^+$ | > | $HCO_3^-$ | > | $Mg^{2+}$ | > | $K^+$ | > | $NO_3^-$ |
| 4 | $SO_4^{2-}$ | $\geq$ | $Cl^-$ | >> | $Ca^{2+}$ | > | $Na^+$ | > | $HCO_3^-$ | >> | $Mg^{2+}$ | > | $K^+$ | > | $NO_3^-$ |

The chloride abundance in clusters 3 and 4 indicates a possible zone of seawater intrusion. Still, it may also result from clay dissolution, anhydrite from sulfide leaching, sewage infiltration, agricultural inputs such as sulfate, and ammonium return of irrigation [18,23]. The spatial distribution of these groups covers most of the aquifer, which indicates a double source of sulfates as irrigation return and seawater intrusion.

The primary domain of cations is $Na^+ > Ca^{2+} >> Mg^{2+}$, and in anions: $SO_4^{2-} >> Cl^- > HCO_3^-$ (i.e., calcium sulfate waters). However, the occurrence of more chlorinated sodium samples is observed in clusters 3 and 4. The distribution of group 4 ($Cl^- >> Na^+$) towards the coastline confirms that a large volume of saltwater intrusion is estimated at the SW of the aquifer [26].

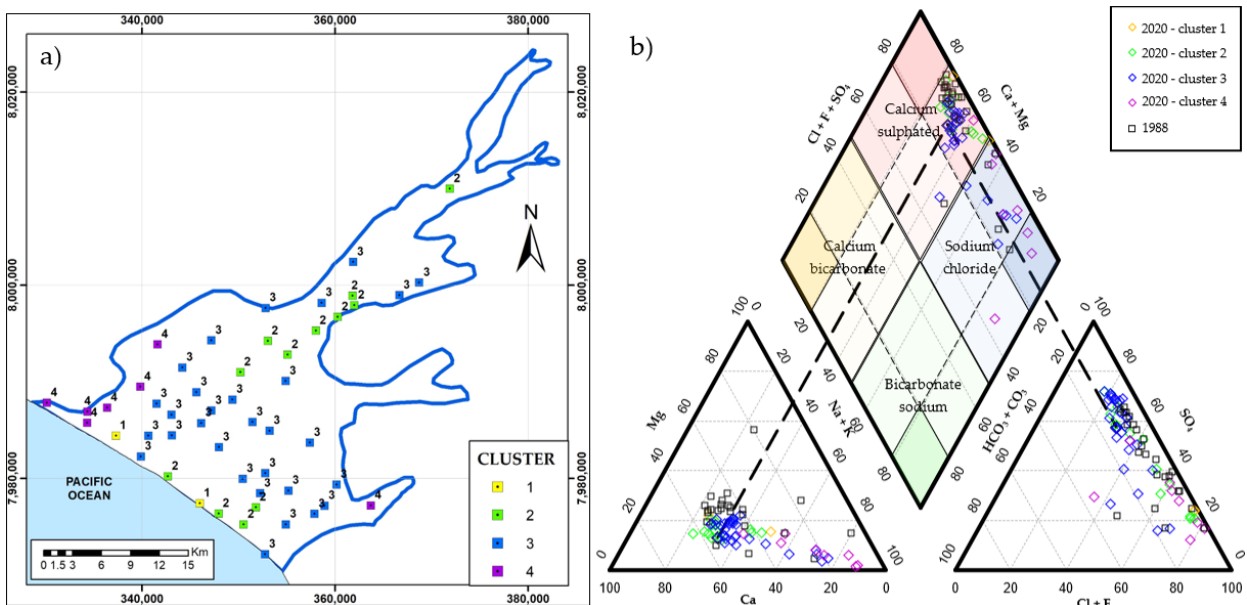

**Figure 7.** (**a**) Spatial distribution of four clusters, obtained from the grouping of 53 groundwaters sampled in 2020. (**b**) Piper diagram for samples from the years 2020 and 1988.

For clusters 1 and 2, the water samples close to the coastline experience $Cl^-$–$SO_4^{2-}$ ion exchange (Figure 7b). These waters contain the highest TDS of the entire sampling network and are influenced by very salty water as a product of seawater intrusion [23,26,32]. However, cluster 2 ($SO_4^{2-} \geq Cl^- >> Ca^{2+}$) waters reappear in the middle part of the aquifer, with hydrochemical values similar to cluster 3 ($SO_4^{2-} >> Cl^-$). Sulfates predominantly salinize the water of cluster 2 due to the SW underground flow that recharges the aquifer [20]. The spatial distribution of water of cluster 3 covers most of the aquifer. There were two sources of sulfate: sewage infiltration and agricultural fertilizer inputs (i.e., irrigation water infiltration) [14,15].

### 4.3.2. Factorial Analysis

Factor analysis permits the identification of the correlations between the physico-chemical components of the groundwater samples. Table 4 presents the three independent factors, classified as strong, moderate, and weak [37], corresponding to values (strong, moderate, and weak correlation) as >0.75, 0.75–0.50, and 0.50–0.30, respectively.

**Table 4.** Values of total variance for the parameters in each factor. Var stands for variance and Acum for accumulative percentage variance.

|  | FA1 | FA2 | FA3 | Communality |
|---|---|---|---|---|
| **EC** | 0.676 | 0.124 | 0.282 | 0.551 |
| **pH** | −0.431 | 0.423 | −0.591 | 0.715 |
| **TDS** | 0.969 | 0.079 | 0.219 | 0.993 |
| **Ca$^{2+}$** | 0.895 | −0.085 | 0.319 | 0.910 |
| **Mg$^{2+}$** | 0.963 | −0.113 | 0.134 | 0.957 |
| **Na$^+$** | 0.766 | 0.301 | 0.116 | 0.690 |
| **K$^+$** | 0.896 | 0.135 | −0.013 | 0.821 |
| **HCO$_3^-$** | 0.064 | −0.039 | 0.702 | 0.499 |
| **SO$_4^{2-}$** | 0.736 | −0.358 | 0.165 | 0.696 |
| **Cl$^-$** | 0.925 | 0.176 | 0.187 | 0.922 |
| **NO$_3^-$** | 0.260 | 0.028 | 0.725 | 0.594 |
| **F$^-$** | 0.204 | 0.794 | 0.084 | 0.678 |
| **$\delta^{18}$O** | −0.035 | 0.971 | −0.044 | 0.946 |
| **$\delta^2$H** | 0.062 | 0.964 | −0.178 | 0.964 |
| **Var** | 6.213 | 2.991 | 1.733 | |
| **%Var** | 0.444 | 0.214 | 0.124 | |
| **Accum** | 0.444 | 0.657 | 0.781 | |

### 4.4. Conceptual Model for Seawater Intrusion in Groundwater Salinization

The water mixing model water delimited the variability of the chemical concentration in the mixture of seawater and freshwater [38]. The La Yarada aquifer groundwaters were clustered in four groups. The factorial analysis converted the 14 parametric chemical and isotopic data into three critical factors: FA1, FA2, and FA3. Figure 8 shows the spatial variation of the first two factors, explaining 44.38% and 21.37% of the total variance (Table 4). The spatial distribution of the chemical signature of the four clusters is also plotted in Figure 8. The factor FA1 is characterized by a positive correlation of $Cl^-$, TDS, $Mg^{2+}$, $Ca^{2+}$, $K^+$, and $Na^+$ and moderately by $SO_4^{2-}$ and EC. FA1 was named the "seawater chemical front", because it represents the primary ions of seawater. Figure 8a shows the spatial distribution of the factor scores. The high-value regions represent high salinization regions ($Cl^-$, TDS, $Mg^{2+}$, $Ca^{2+}$, $K^+$, and $Na^+$), and the low-value regions denote the discharge regions for these chemical species.

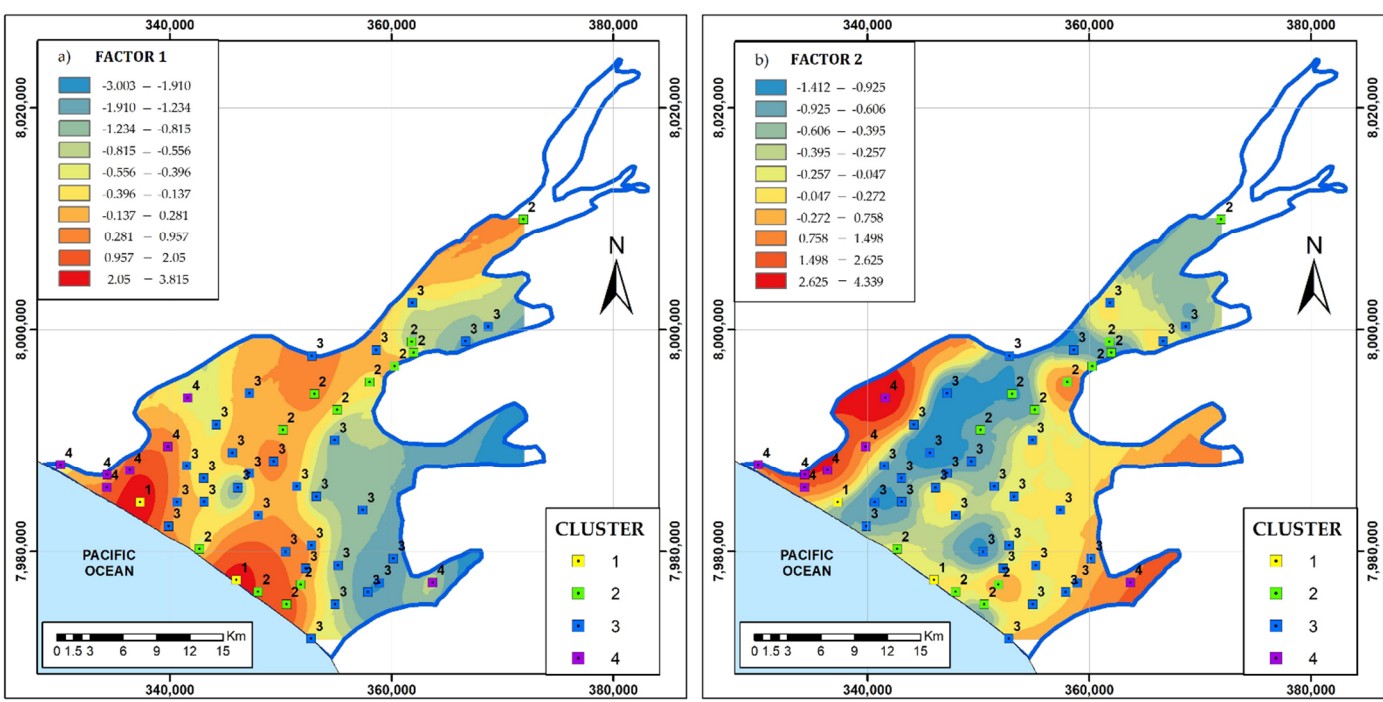

**Figure 8.** Spatial distribution of factors: (**a**) FA1 (seawater chemical front) and (**b**) FA2 (seawater isotopic front). The clusters were drawn from Figure 7 to compare the different multivariate approaches.

The FA2 factor characterized the positive correlation of isotopes ($\delta^2H$, $\delta^{18}O$) and ion $F^-$ (Table 4); therefore, it was named the "seawater isotopic front". Figure 8b represents the spatial distribution of FA2 scores.

The spatial distributions of FA1 and FA2 suggest two processes of seawater front movement: dispersion (diffusion) of chemical elements and different types of water mixing, respectively. At the edge of the La Yarada aquifer, the water head was relatively low, permitting seawater and freshwater mixing (Figure 8b). Along the sea-land boundary, the water head of the La Yarada aquifer was relatively high, avoiding seawater and freshwater mixing; however, the chemical species were migrating from the seawater to the groundwater due to the diffusion processes.

The FA3 factor explains 12.38% of the total variance. Moderate loads of $HCO_3^-$ and $NO_3^-$ characterized with a negative correlation of pH and $NO_3^-$. It means an entry of municipal wastewater infiltrated from the forest in the center of the aquifer since the forest was irrigated with secondary treated wastewater. The $HCO_3^-$ may also be originated from the dissolution of carbonates. The negative correlation between $HCO_3^-$ and pH would indicate the little significant impact of pH in the dissolution of carbonates.

There is a good relationship between clusters and factors. The data points corresponding to cluster 4 are located in the region where the FA2 seawater isotopic front process dominates. Similarly, cluster 4 represents the process corresponding to the FA1 seawater chemical front.

### 4.4.1. Sama Basin Underground Connection

The Sama basin has the $\delta^{18}O$ values ($-10$ to $-12‰$). The multitemporal iso-value maps in Figure 5 show the variation from $-11‰$ to $-10‰$ in 32 years (i.e., isolines in the SE region of the aquifer). This theory is reinforced when observing the distribution of cluster 2 (Figure 7) in superposition to the higher scores in FA2 (Figure 8b). It may be inferred that there is an underground flow to recharge the wells to the east of the aquifer. Under the above postulation, the isotopic composition of the water, once in the aquifer, does not change quickly, except for mixing with another source of different isotopic values [35].

### 4.4.2. Rock Dissolution

The $\delta^{18}O$ value tends towards more positive values that are influenced by the isotopic exchange in the water-rock reactions, favored by the low slope (3–5%) in the aquifer [39,40]. The soil's primary source of fluoride, emphasized by the factor FA2 and cluster 4, is clay minerals [41]. Similarly, paleochannels [39] were identified in piezometric records of the aquifer, formed of silt and clay banks. These paleochannels were formed due to weathering, and the leaching process occurred during the groundwater flow. When fluoride-rich minerals are in contact with highly alkaline water, they release fluoride into groundwater through hydrolysis, replacing the hydroxyl ion ($OH^-$) [42]. It is worth highlighting the considerable positive correlation between fluoride and pH in factor FA2 (Table 4). The rock dissolution process is not only present in the aquifer strata but throughout the Caplina basin of the sulfated calcium family from the oxidation of sulfides due to the hydrothermal alteration in the mineral occurrences of the intrusive Yarabamba and Challaviento [40].

### 4.5. Seawater Salinization Processes

Figure 9 presents the conceptual model of the negative effect on the hydraulic barriers [32] and geometry of the freshwater–seawater contact after 13 years of operation of extraction wells [33].

The $\gamma$-factor for parameters ($Cl^-_{sub}$, $Na^+_{sub}$, and $\delta^{18}O_{sub}$) was calculated by Equation (1), considering the distance as their position perpendicular to the coastline in the NW direction. The factor represents the fraction of seawater origin in the groundwater samples. The data were fitted in the logarithmic functions (Figure 9a). Despite having a positive correlation, the samples are very dispersed to the trend consequence of seawater salinization anomalies. In contrast, the rest of the samples with $\gamma = 0$–$0.05$ depends on the aquifer's hydrogeological and paleo-hydrogeological regimes. In this sense, in Figure 9a, emphasis was placed on relating the anomalies of 1988 and 2020 with factors $\gamma = 0.20$–$0.30$, configuring boundary lines of the facies with the highest seawater proportion (Figure 9b).

A typical mixing interface was obtained for the data of the year 1988. For the year 2020, at 11 km, samples that differ from the behavior of a salinized interface are distributed, generating a behavior similar to that obtained after applying hydraulic barriers [32]. Based on these already established models of the effects of negative hydraulic barriers [31,32], a 3D model was generated to visualize the "pseudo hydraulic barriers" (Figure 9b).

The extensive exploitation would drive this behavior at 11 km from the coastline that overlaps in the range of exploited volumes (Figure 9c). This anomaly maintains the isotopic signature in the aquifer (Figure 3) since exploitation is continuous every month of the year. Therefore, they do not allow the majority salinization of $Cl^-$ and $Ca^{2+}$ in more than 30% of the aquifer. The graph of the $\gamma$ fraction versus littoral distance (Figure 9a) appears to be a process called saltwater up-coning [43] as a result of the intense pumping of water in the area (Figure 9c).

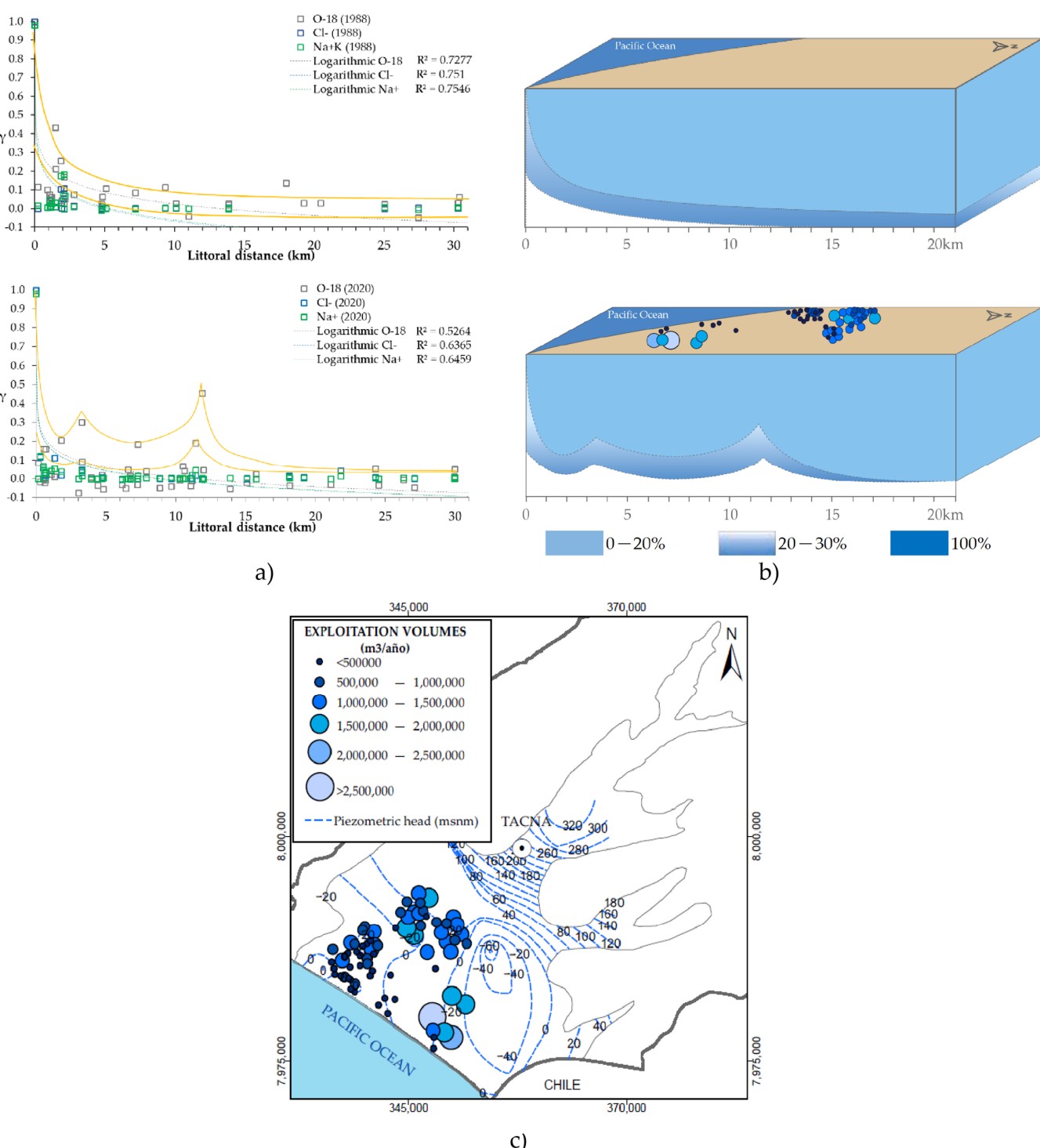

**Figure 9.** (**a**) Logarithmic correlation of the proportion factor (γ) versus littoral distance along with anomalous distribution configuration. (**b**) 3D model of the interface (γ = 0.20–0.30) represented in percentages. (**c**) Map of exploitation volume in the La Yarada aquifer and piezometric levels for 2018.

## 5. Conclusions

The multivariate analysis and chemical–isotopic trends studying different types of water in the semi-arid Tacna, Peru, to investigate seawater intrusion and hydrogeological processes affecting water quality are summarized in the following outcomes:

1.  No change in the chemical and isotopic analysis of groundwater of two samplings, performed in the dry (August 2020) and wet (November 2020), indicate practically no direct influence of precipitation on the La Yarada aquifer characteristics.
2.  The hydrochemical and isotopic mixing model supports the formation of a wedge with 20% seawater intrusion. The heterogeneity of piezometric map isolines corroborates the wedge formation associated with the groundwater movement.
3.  The factor analysis reduces the chemical and isotopic parameters into the FA1 seawater chemical front and the FA2 seawater isotopic front.
4.  In the La Yarada aquifer, the seawater and freshwater mixing is dominant at the SW of the aquifer; however, the chemical species migration is along the sea-land boundary. The cluster 4 samples are in the region corresponding to the isotopic mixing process represented by the FA2, while cluster 4 describes the chemical diffusion process represented by the FA2.
5.  The intrusion of the mixing zone into groundwater behaves differently in the unconfined zone due to the variability of exploitation volumes. In the area proximal to the coastline, the extracted volumes do not exceed 0.5 $Hm^3$. In contrast, more wells than those registered exist to pump 2.5 $Hm^3$ water from the Caplina aquifers nearby the wedge formation.

**Author Contributions:** Conceptualization, E.P.-V., M.P.V. and A.V.; software, S.C. and A.V.; validation, E.P.-V.; formal analysis, E.C. and M.C.; writing—original draft preparation, A.V.; writing—review and editing, J.A.T.-M., A.M. and J.M.; supervision; project administration, E.P.-V. All authors have read and agreed to the published version of the manuscript.

**Funding:** Universidad Nacional Jorge Basadre Grohmann, Tacna, Perú.

**Institutional Review Board Statement:** Not applicable.

**Informed Consent Statement:** Not applicable.

**Data Availability Statement:** Not applicable, part of the data corresponds to reports from Peruvian public institutions that under agreement provided such information with academic purposes, but the majority of data is in Table 2.

**Acknowledgments:** This work was developed within the framework of the research project, "Integration of hydrodynamic, hydrochemical and isotopic methods to specify the operation and sustainable management of the La Yarada aquifer, Tacna, Peru" founded by "Fondos de canon, sobrecanon y regalías mineras de la Universidad Nacional Jorge Basadre Grohmann, Tacna, Perú". The constructive comments of two anonymous reviewers helped to improve the content and presentation of this manuscript.

**Conflicts of Interest:** The authors declare no conflict of interest.

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
