# Peer review of "Hydrodynamics, Hydrochemistry, and Stable Isotope Geochemistry to Assess Temporal Behavior of Seawater Intrusion in the La Yarada Aquifer in the Vicinity of Atacama Desert, Tacna, Peru"

_water, doi:10.3390/w13223161_

Round 1
Reviewer 1 Report
Review of the paper "Hydrodynamics, hydrochemistry, and stable isotope geochemistry to assess temporal behavior of marine intrusion in the coastal aquifer in the vicinity of Atacama Desert, Tacna, Peru” by Vera et al.
This paper presents an interesting case study for the application of multivariate statistic and geochemical trends in water to investigate the effect of mixing with saltwater and the other hydrogeological processes affecting water quality. The main findings make this paper an interesting contribution and the aim of the study make it suitable for the publication in “water”. However, there are several major issues (especially regarding the description of methodology and the quality of presentation of results) which needs to be improved.
Firstly, abstract needs to be revised. A sentence including clearly the main aim of the study needs to be added, as well as a final sentence summarizing the main repercussions of the findings in the study.
Moreover, analytical methods need to be included in the main text, as well as QA/QC protocols in sampling and processing of water samples. Which methods were applied for the major ions analysis? And for the isotopes? Which pre-treatment of samples were performed? This information needs to be added.
Then, the section of results and discussion needs to be reframed to better highlight the main findings of the study. Graphical appearance and order of figures should be reorganized to better indicate the trends observed by the authors.
In figure 4, the snow values is not easily readable. Please change the symbol. Moreover, please change the format of the regression equations for the different lines, since it is not understandable to which they refer.
It also seems that figure 6 is not very informative considering the short descriptions in the paper. It should be reasonable to move it as supplementary material. Moreover, please revise the font and the appereance of the graphs in this figure, since it is not easy to read as it is. I also suggest to name the different panels to better highlight them. Please also indicate the original values of the different sampling sites as points in the graph. Moreover, what is superficie in the phreatic level panels? Is it the ground level?
In figure 9 It can be good to overlap the clusters indicated in figure 8 to these interpolation maps, to validate if the different multivariate approaches helped in understanding the different water sources. Therefore, I suggest to move the figure 11 d here and make the same type of picture for factor 2. Moreover, in in figure 11d a legend of clusters is missing. I would also suggest to the authors to understand if the clustering changes in the different seasons, to explore possible temporal effects in the different sampling points (see e.g., Binda et al., 2020 https://doi.org/10.1007/s10653-019-00405-4 or Gabrielli et al., 2008 https://doi.org/10.1016/j.chemosphere.2008.04.076 ).
The text in lines 267-323 needs to be re-arranged. It should be more intuitive to create a section discussing all the mechanisms of elements dissolution in water and the mixing (e.g., with the title "conceptual model"), and then discuss as sub-sections the different processes affecting the aquifer.
Regarding table 3, I think that a graph (likely a piper diagram) can be more informative than this table.
Moreover, a general revisions of English language and misspellings need to be performed (e.g., in line 60 semicolon instead of dot; remove the parenthesis in line 92, rephrase in lines 132-133 and in lines 140-144, “high scattering” instead of “high scattered” in line 138, etc.).
Finally, there are some minor issues which need to be revised:
- Figure 1: show bigger font for legend and the labels in the left-down panel. Moreover, I suggest to add a reference (e.g., a), b), c) ) for the different panels.
- Line 96-97: Please add a reference or the seasonal precipitation data when indicating the dry and wet season
- Equation 1: Please define Xmix
- Line 170-172: Please better define how can these phenomena affect the change in isotopic composition.
- Line 321: it unclear why the figure showing the factor analysis interpolation and the clusters highlight the Cl-Ca hydrochemical signature. Please add more details.
- Line 327: Was it done through factor analysis or through cluster analysis?
Author Response
Reply to reviewer 1 comments: Comment is in black, and the respective reply is in Red
Hydrodynamics, hydrochemistry, and stable isotope geochemistry to assess temporal behavior of seawater intrusion in the La Yarada aquifer in the vicinity of Atacama Desert, Tacna, Peru
The revised manuscript is entirely rewritten; therefore, we did not submit a marked copy of the manuscript.
REVIEWER 1
This water quality. The main findings make this paper an interesting contribution and the aim of the study make it suitable for the publication in "water". However, there are several major issues (especially regarding paper presents an interesting case study for the application of multivariate statistic and geochemical trends in water to investigate the effect of mixing with saltwater and the other hydrogeological processes affecting the description of methodology and the quality of presentation of results) which needs to be improved.
Firstly, abstract needs to be revised. A sentence including clearly the main aim of the study needs to be added, as well as a final sentence summarizing the main repercussions of the findings in the study.
We rewrote the abstract with the main aim of the study.
Moreover, analytical methods need to be included in the main text, as well as QA/QC protocols in sampling and processing of water samples. Which methods were applied for the major ions analysis? And for the isotopes? Which pre-treatment of samples were performed? This information needs to be added.
We added a section, "3.2 Analysis and Data Quality," to present the analytical procedure of each type of measurement.
Then, the section of results and discussion needs to be reframed to better highlight the main findings of the study. Graphical appearance and order of figures should be reorganized to better indicate the trends observed by the authors.
We restructured the manuscript content, including the figures to highlight the finding of the manuscript better.
In figure 4, the snow values is not easily readable. Please change the symbol. Moreover, please change the format of the regression equations for the different lines, since it is not understandable to which they refer.
We changed the symbol for the snow sample and added the text "snow" near the location of the snow sample in Figures 1 and 4.
It also seems that figure 6 is not very informative considering the short descriptions in the paper. It should be reasonable to move it as supplementary material. Moreover, please revise the font and the appereance of the graphs in this Figure, since it is not easy to read as it is. I also suggest to name the different panels to better highlight them. Please also indicate the original values of the different sampling sites as points in the graph. Moreover, what is superficie in the phreatic level panels? Is it the ground level?
Figure 6 is removed and included a short description in the paper. The fonts were rechecked in the whole manuscript to fulfill the journal requirements
In figure 9 It can be good to overlap the clusters indicated in figure 8 to these interpolation maps, to validate if the different multivariate approaches helped in understanding the different water sources. Therefore, I suggest to move the Figure 11 d here and make the same type of picture for factor 2. Moreover, in in Figure 11d a legend of clusters is missing.
Thank you for the suggestion. We added the cluster in Figure 9, now Figure 8. Figure 11d was removed in the revised version of the manuscript. There is a good relationship between the two approaches. It is described in the manuscript.
I would also suggest to the authors to understand if the clustering changes in the different seasons, to explore possible temporal effects in the different sampling points (see e.g., Binda et al., 2020 https://doi.org/10.1007/s10653-019-00405-4 or Gabrielli et al., 2008 https://doi.org/10.1016/j.chemosphere.2008.04.076 ).
Thanks for suggesting the temporal effects in cluster analysis. We read the suggested references and included their findings in the manuscript. We aimed the cluster analysis for groundwater in the La Yarada aquifer. There were no observed variations in these waters' chemical and isotopic compositions, so there was no temporal variation effect on the cluster analysis.
The text in lines 267-323 needs to be re-arranged. It should be more intuitive to create a section discussing all the mechanisms of elements dissolution in water and the mixing (e.g., with the title "conceptual model"), and then discuss as sub-sections the different processes affecting the aquifer.
The section, "Results and Discussion," was rewritten entirely with considering the suggestions.
Regarding table 3, I think that a graph (likely a piper diagram) can be more informative than this table.
Table 3 defines the strategy of clustering. Similarly, the Piper diagram is added for better clarity.
Moreover, a general revisions of English language and misspellings need to be performed (e.g., in line 60 semicolon instead of dot; remove the parenthesis in line 92, rephrase in lines 132-133 and in lines 140-144, "high scattering" instead of "high scattered" in line 138, etc.).
The suggestions were included in the manuscript.
Finally, there are some minor issues which need to be revised:
- Figure 1: show bigger font for legend and the labels in the left-down panel. Moreover, I suggest to add a reference (e.g., a), b), c) ) for the different panels.
The legend fonts of Figure 1 were corrected, and the Figures were numbered as a, b, and c.
- Line 96-97: Please add a reference or the seasonal precipitation data when indicating the dry and wet season
The wet season is generally October to December (January] in the southern hemisphere and the dry season in June -September. We added two references.
- Equation 1: Please define Xmix
The subscript (mix) indicates the mixture component (i.e., the La Yarada aquifer) groundwater. Now, it has been added to the manuscript.
- Line 170-172: Please better define how can these phenomena affect the change in isotopic composition.
Conclusions were rewritten.
- Line 321: it unclear why the Figure showing the factor analysis interpolation and the clusters highlight the Cl-Ca hydrochemical signature. Please add more details.
There is a good relationship between the two approaches, cluster and factor analyses, illustrated in Figure 8 and explained in the text.
- Line 327: Was it done through factor analysis or through cluster analysis?
Conclusions were rewritten, including the relations between the cluster and factor analyses.

Reviewer 2 Report
General comments:
*I recommend, you include in the title the name of the aquifer
*1. Introduction:
-I recommend you to re-read the introduction and re-write it.
-you must write about some different cases that they have use the same methods like you, and they works it and previous researchers in the same location. You must explain, why you use these methods and not others, prove with other papers.
-You must delete some paragraphs, which you write about location, climatic characteristics and socioeconomics. I recommend you to write a new sections inside 2.Materials and methods (1**):
- Site description: this section must include location and socioeconomic description
- Hydrogeological settings: you must describe the hydrogeological characteristics (materials, aquifer geometry, relationships with other aquifers, seawater intrusion, water budget, …)
* Sometimes, you use “seawater” and sometimes “marine water”. I recommend you to use always the same word
*2. Materials and methods
-see (1**)
-I recommend you to include a new section “monitoring network and sampling”, and explain that in deep
- Each method that you use to obtain results, you must explain it, for example cluster, …
- 2.3 I need some reference about equation 1
- Figure 10. You must change the location, it is better to put in the section 2, I recommend to you to do a new section inside “2. Materials and methods,” about “seawater intrusion conceptual model (2**)” and explain inside upconing process too.
*4. Conclusions.
-I don’t understand the difference between “salinization factor2 and “marina salinization factor”, please explain it.
-You must re-think and re-write the conclusions because, between line 325 and 332, and 334 to 337, they are results not conclusions.
Specific comments:
*Abstract: Check the letter sizes, there are two different.
*Figure 2 (a, b, c, d) is not relevant , I recommend you to explain the situation in the manuscript main body, and the figure to put in the annex
*Figure 3: Legend is so small
*It is not clearly if the figure 3 are you research or from other researches, so it is important to note that.
*Line 136, what is Craig diagram? You must explain in section 2
*Line 141, masl. The first time, you write it, you must explain what it is
*Line: 145 “the isotopic…Basin”, it is conclusions not results.
*Figure 4: the figure caption is not correct. You do not explain the results inside it
*Figure 5. The legend is to small, and I recommend to change the colours of the legend, blue for all is a bit confusing
*Table 2: You must write the upper index of the anions and cations, and explain CE, ph, TDS, …
*Line 209, it is conclusions not results
*Line 222, it is conclusions not results
*table 4, you must explain it
*figure 8: rivers and boundaries some colour, blue? Please change it
*Line 250 till 254. It is not you results, it is a explanation of you result, so you must write that in section 2
*Figure 10. You must change the location, it is better to put in the section 2, I recommend to you to do a new section inside “2. Materials and methods,” about “seawater intrusion conceptual model (2**)”
Author Response
Reply to reviewer 2 comments: Comment is in black, and the respective reply is in Red
Hydrodynamics, hydrochemistry, and stable isotope geochemistry to assess temporal behavior of seawater intrusion in the La Yarada aquifer in the vicinity of Atacama Desert, Tacna, Peru
The revised manuscript is entirely rewritten; therefore, we did not submit a marked copy of the manuscript.
REVIEWER 2
*I recommend, you include in the title the name of the aquifer
The name of the aquifer, La Yarada, was included in the title.
*1. Introduction:
-I recommend you to re-read the Introduction and rewrite it.
-you must write about some different cases that they have use the same methods like you, and they works it and previous researchers in the same location. You must explain, why you use these methods and not others, prove with other papers.
The Introduction was rewritten according to your suggestions.
-You must delete some paragraphs, which you write about location, climatic characteristics and socioeconomics. I recommend you to write a new sections inside 2.Materials and methods (1**):
- Site description: this section must include location and socioeconomic description
- Hydrogeological settings: you must describe the hydrogeological characteristics (materials, aquifer geometry, relationships with other aquifers, seawater intrusion, water budget, …)
The section, Study Area, was added to describe the "Location and Climate" and "Hydrogeological Settings". Similarly, the section, Materials and Methods, was rewritten to include "Monitoring Network and Sampling" and "Analysis and Data Quality".
* Sometimes, you use "seawater" and sometimes "marine water". I recommend you to use always the same word
Now, "seawater" is used everywhere for "marine water".
*2. Materials and methods
-see (1**)
-I recommend you to include a new section "monitoring network and sampling", and explain that in deep
- Each method that you use to obtain results, you must explain it, for example cluster, …
The materials and Methods section was rewritten to include "Monitoring Network and Sampling" and "Analysis and Data Quality". The cluster analysis is based on the computation of a Euclidean distance dissimilarity measure
- 2.3 I need some reference about equation 1
A reference for equation 1 is cited in the manuscript.
- Figure 10. You must change the location, it is better to put in the section 2, I recommend to you to do a new section inside "2. Materials and methods," about "seawater intrusion conceptual model (2**)" and explain inside upcoming process too.
Figure 10 is moved to the recommended section.
*4. Conclusions.
-I don't understand the difference between "salinization factor2 and "marina salinization factor", please explain it.
We wrote better names for the factors, depending on the parameters. For better clarity, factor FA1 is named as "seawater chemical front" and FA2 as "seawater isotopic front". The relationship between the two approaches is explained in Figure 8 and in the text.
-You must re-think and rewrite the conclusions because, between line 325 and 332, and 334 to 337, they are results not conclusions.
The conclusions were rewritten.
Specific comments:
*Abstract: Check the letter sizes, there are two different.
The font size was corrected according to the template.
*Figure 2 (a, b, c, d) is not relevant , I recommend you to explain the situation in the manuscript main body, and the figure to put in the annex
Figure 2 was removed.
*Figure 3: Legend is so small
Legend was improved to make clear the content of Figure 3.
*It is not clearly if the figure 3 are you research or from other researches, so it is important to note that.
Figure 3 was drawn in this study. Now, we made it clear.
*Line 136, what is Craig diagram? You must explain in section 2
We added the original reference for it.
*Line 141, masl. The first time, you write it, you must explain what it is
Now, masl was defined as meters above sea level.
*Line: 145 "the isotopic…Basin", it is conclusions not results.
Conclusions were rewritten.
*Figure 4: the figure caption is not correct. You do not explain the results inside it
The fitted lines are made clear in the figure.
*Figure 5. The legend is to small, and I recommend to change the colours of the legend, blue for all is a bit confusing
We changed the color of sample points to avoid confusion.
*Table 2: You must write the upper index of the anions and cations, and explain CE, ph, TDS, …
The meaning of EC – Electrical conductivity, pH, and TDS- total dissolved solids is included in the manuscript.
*Line 209, it is conclusions not results
*Line 222, it is conclusions not results
*table 4, you must explain it
*figure 8: rivers and boundaries some colour, blue? Please change it
We changed the color of sample points to avoid confusion.
*Line 250 till 254. It is not you results, it is a explanation of you result, so you must write that in section 2
The lines are removed.
*Figure 10. You must change the location, it is better to put in the section 2, I recommend to you to do a new section inside "2. Materials and methods," about "seawater intrusion conceptual model (2**)"
Figure 10 is moved to section 3.
Round 2
Reviewer 1 Report
The revised version of the paper "Hydrodynamics, hydrochemistry, and stable isotope geochemistry to assess temporal behavior of seawater intrusion in the La Yarada aquifer in the vicinity of Atacama Desert, Tacna, Peru" actually improved its quality and redeability, making the paper better suitable for publication.
There are still few minor issues to fix in my opinion, and they are listed below:
-line 22: "analysis of water" instead of "analysis water";
-line 31: "La aquifer". Is that "La Yarada"?
-line 64: remove the parenthesis before 2H
-line 119: "....following the method ISO 17294-2" instead of "...ISO 17294-2"
-line 126: "written under this project" What is the meaning of this project? Please rephrase
-line 229-231: unclear sentence, please rephrase;
-line 269: "Factor analysis was to obtain the correlations between..." I gues there is some typo, this sentence is not clear to me.
-lines 328-332: I suggest to rephrase this paragraph, text is too fragmented, making its reading difficult.
Author Response
There are still a few minor issues to fix, in my opinion, and they are listed below:
-line 22: "analysis of water" instead of "analysis water";
Corrected
-line 31: "La aquifer". Is that "La Yarada"?
Yes, it was corrected.
-line 64: remove the parenthesis before 2H
The parenthesis was removed.
-line 119: "....following the method ISO 17294-2" instead of "...ISO 17294-2"
It was corrected.
-line 126: "written under this project" What is the meaning of this project? Please rephrase.
The computer program was written during the present study, and it was not published yet. It was corrected.
-line 229-231: unclear sentence, please rephrase;
The lines were rewritten to clarify the content.
-line 269: "Factor analysis was to obtain the correlations between..." I gues there is some typo, this sentence is not clear to me.
The sentence was rephrased.
-lines 328-332: I suggest to rephrase this paragraph, text is too fragmented, making its reading difficult.
The lines were rewritten to clarify the content.
Reviewer 2 Report
Specific comments:
Section 3.1: line 108: “100 sampling points…” It would be interesting to explain if these 100 sampling points, a general point of view, are form agricultural, industry or urban sector.
In line 131, 159 and 185 you say “Caplina aquifer”, but I think that you wat to say Caplina basin, is it a mistake? Please check it!
Figure 4. line 173 “ the isotopic…basins”. It a result, so you must write that in the appropriate section, not in the figure caption.
Author Response
Specific comments:
Section 3.1: line 108: "100 sampling points…" It would be interesting to explain if these 100 sampling points, a general point of view, are from the agricultural, industry, or urban sector.
Of the total samples, 76 are for agriculture use, 9 for domestic use, 5 for cattle-ranch use, 3 for urban use, 1 for industrial use, and 6 for no use due to high contamination.
In line 131, 159 and 185 you say "Caplina aquifer", but I think that you want to say Caplina basin, is it a mistake? Please check it!
Yes, it was Caplina basin, corrected.
Figure 4. line 173 "the isotopic…basins". It is a result, so you must write that in the appropriate section, not in the figure caption.
Thank you. The sentence is removed from the figure caption and included only in the section of results.